# Stage IV Gastric Cancer: The Surgical Perspective of the Italian Research Group on Gastric Cancer

**DOI:** 10.3390/cancers12010158

**Published:** 2020-01-09

**Authors:** Silvia Ministrini, Maria Bencivenga, Leonardo Solaini, Chiara Cipollari, Silvia Sofia, Elisabetta Marino, Alessia d’Ignazio, Beatrice Molteni, Gianni Mura, Daniele Marrelli, Maurizio Degiuli, Annibale Donini, Franco Roviello, Giovanni de Manzoni, Paolo Morgagni, Guido A. M. Tiberio

**Affiliations:** 1Clinica Chirurgica, Università di Brescia, 25100 Brescia, Italy; silvia.ministrini@hotmail.it (S.M.); Beatrice.molteni@hotmail.com (B.M.); 2Chirurgia Generale, Università di Verona, 37100 Verona, Italy; mariabenci@hotmail.it (M.B.); chiaracipollari@hotmail.it (C.C.); Giovanni.demanzoni@univr.it (G.d.M.); 3Chirurgia Generale, Ospedale di Forlì, Università di Bologna, 47121 Forlì, Italy; leonardo.solaini2@unibo.it (L.S.); morgagni2002@libero.it (P.M.); 4Chirurgia Generale, Università di Torino, 10121 Torino, Italy; sofiaasilvia@gmail.com (S.S.); maurizio.degiuli@unito.it (M.D.); 5Chirurgia Generale, Università di Perugia, 06121 Perugia, Italy; elisabetta.marino1986@gmail.com (E.M.); annibale.donini@unipg.it (A.D.); 6Chirurgia Oncologica, Università di Siena, 53100 Siena, Italy; docalessia89@gmail.com (A.d.); daniele.marrelli@unisi.it (D.M.); franco.roviello@unisi.it (F.R.); 7Chirurgia Generale, Ospedale di Arezzo, 52100 Arezzo, Italy; gianmura@gmail.com

**Keywords:** gastric cancer, metastasis, surgery, moltimodal treatment, prognostic factors

## Abstract

*Background/Aim:* This work explored the prognostic role of curative versus non-curative surgery, the prognostic value of the various localizations of metastatic disease, and the possibility of identifying patients to be submitted to aggressive therapies. *Patients and Methods:* Retrospective chart review of stage IV patients operated on in our institutions. *Results:* Two hundred and eighty-two patients were considered; 73.4% had a single metastatic presentation. In 117 cases, a curative (R0) resection of primary and metastases was possible; 75 received a R1 resection and 90 a palliative R2 gastrectomy. Surgery was integrated with chemotherapy in multiple forms: conversion therapy, HIPEC, neo-adjuvant and adjuvant treatment. Median overall survival (OS) of the entire cohort was 10.9 months, with 14 months for the R0 subgroup. There was no correlation between metastasis site and survival. At multivariate analysis, several variables associated with the lymphatic sphere showed prognostic value, as well as tumor histology and the curativity of the surgical procedure, with a worse prognosis associated with a low number of resected nodes, D1 lymphectomy, pN3, non-intestinal histology, and R+ surgery. Considering the subgroup of R0 patients, the variables pT, pN and D displayed an independent prognostic role with a cumulative effect, showing that patients with no more than 1 risk factor can reach a median survival of 33 months. *Conclusions:* Our data show that the possibility of effective care also exists for Western patients with stage IV gastric cancer.

## 1. Introduction

Gastric cancer is diagnosed at stage IV in 35–55% of cases in Western countries [1], and median disease-specific survival at this stage is approximately 10 months [2], with overall 5-year survival estimated to be 3–5% [3,4]. The results of the REGATTA Trial [5] indicate that surgery should be avoided in stage IV gastric cancer. Indeed, palliative gastrectomy is much more invasive than chemotherapy and achieves similar survival results. However, in their trial, Fujitani and colleagues did not consider the possibility of a complete surgical resection of both gastric cancer and metastases, limiting their comparison between chemotherapy alone versus gastrectomy plus chemotherapy in the metastatic setting. The recent literature [6,7,8,9,10,11,12,13,14,15] contrasts with the conclusion of the REGATTA Trial, showing that an integrated multidisciplinary approach including chemotherapy and surgery may offer, at least to a selected subgroup of patients, unexpected results, in particular when a radical (R0) resection can be achieved both on gastric primary and metastases. Although the power of these studies is limited by their retrospective nature, they cannot be ignored, particularly in the light of results achieved by conversion therapy. As initially reported by Yoshida et al. [16] and confirmed on a larger scale by the CONVO GC-1 study, astonishing survival performances can even be achieved in categories 3 and 4 of Yoshida’s classification [17].

As one third of patients are diagnosed with stage IV disease, it is important to establish therapeutic standards. Unfortunately, it is extremely difficult to conduct randomized trials in this heterogeneous group of patients, and conclusive results may not be achieved for a long time.

Extending our interest beyond the subset of patients affected by gastric cancer with hepatic metastases, for whom we actively endorsed the role of surgery [6,9,10,11], we investigated the role of surgery in our entire cohort of metastatic patients. Our research had three main purposes: to compare the survival performance of palliative gastrectomy and curative resection of both gastric cancer and metastases; to evaluate whether the different presentations of the metastatic disease have the same prognostic value, as suggested by the TNM classification; and to explore whether it is possible to recognize patients to be submitted to aggressive multimodal management.

## 2. Materials and Methods

Retrospectively, we reviewed data regarding patients submitted to surgical treatment of stage IV gastric cancer from January 2011 to June 2018. Data were extrapolated from a prospectively collected multi-centric database, shared by 7 institutions, members of the Italian Research Group on Gastric Cancer. Data were managed according to institutional rules with patient consent. This research was approved by Comitato Etico Romagna (Code 2255, protocol number 8441/2018, permission date 18 November 2018).

Only patients with pathologically confirmed gastric cancer associated with one or more metastatic sites were enrolled in the study.

Pre-operative work-up systematically included total body computed tomography and staging laparoscopy, when indicated.

Chemotherapy schemes varied among the various centers and during the study period; independently of whether administered preoperatively or postoperatively, they included: fluoropyrimidine and cisplatin ± epirubicin ± leucovorin, or the association of oxaliplatin ± taxane ± fluoropyrimidine. The latter was almost systematically employed after the advent of the FLOT regimen [18].

The minimal surgical procedure was a gastrectomy. Completeness of surgical resection was established according to the R0-1-2 criteria, where R0 indicated a curative resection, both on the stomach and on the metastases. R1 indicated microscopic tumor residual on the surgical margin, indifferently of whether on the gastric margin or on the metastasis’ margin. R2 indicated macroscopic metastatic residual. All R2 patients received a palliative gastrectomy.

Five of the 7 participating institutions adopted Hypertermic Intra PEritoneal Chemotherapy (HIPEC) in selected cases. This was performed at the end of the surgical procedure by a closed abdomen technique, using cisplatin and doxorubicin at 41–42 °C for 60 min. HIPEC was only proposed to ECOG 0 patients under 70 years old.

Pathologic data concerning gastric primary and metastases were collected as suggested by the general rules of the JGCA and classified following the 8th AICC-TNM system [19,20]. Follow-up was structured as previously described [21] and stopped in December 2018.

We evaluated whether survival varied according to the different presentations of the metastatic disease and whether it was influenced by one or more of the considered candidate prognostic factors related to patients, gastric cancer, metastasis, and treatment.

### Statistical Analysis

Descriptive statistics are presented as median and interquartile range (IQR, 25%–75%) or confidence interval (CI). Different study groups were compared by means of χ^2^ or *t*-student test for discrete or continuous variables, respectively. Statistical significance was rated at *p* < 0.05. Overall survival (OS) was measured from the date of resection to the date of death or latest follow-up. Survival curves were generated by the Kaplan–Meier method and compared by log-rank test. Variables that resulted to be statistically significant (*p* < 0.05) at univariate analysis were considered for multivariate analysis with a Cox proportional hazards model.

## 3. Results

In this study, 282 patients were considered, 181 males and 101 females. The median age was 67 (IQR: 56–75 years).

Of these, 207 (73.4%) had a single metastatic presentation; this was peritoneal in 126 patients (44.7%), hepatic in 45 (16.0%), into distant lymph nodes in 32 (11.3%), and haematogenous but extrahepatic in 4 (1.4%). Multiple metastatic localizations were present in 75 patients (26.6%).

Preoperative ASA score was 1 in 40 cases (14.2%), 2 in 105 (37.2%) and 3–4 in 91 (32.3%); ASA score was unknown in 46 cases (16.3%). ECOG performance status was 0 in 121 patients (42.9%), ECOG 1–2 in 36 (12.8%), and unknown in the remaining 125 cases (44.3%).

The histotype of gastric cancer was intestinal in 112 patients (39.7%), diffuse in 122 (43.2%), mixed in 41 (14.5%), and classified differently in 7 cases (2.6%).

Clinical staging is detailed in Table 1. It should be noted that in 37.5% of cases (106 patients), the presence of synchronous metastases was assessed at surgical exploration; diagnostic laparoscopy was carried out in 44 cases (15.6%): peritoneal metastases were recognized in 22 cases (50.0%) and peritoneal fluid collection resulted positive (CY1) in 15 (34.1%).

In 115 cases (40.7%), symptoms such as bleeding, occlusion or perforation were present at diagnosis. In 73 cases, symptoms required immediate surgery, sometimes in emergency conditions, and in 66 of them (90.4%) a palliative gastectomy was performed.

In 164 patients (58.1%), gastrectomy was associated with resection of metastases with curative intent.

Forty-five patients (15.9%), initially considered as non-resectable, were submitted to chemotherapy and therefore operated on (conversion surgery group).

Total gastrectomy was performed in 139 cases (49.3%) and subtotal in 143 (50.7%); lymphadenectomy was D1 in 70 cases (24.8%), and D2 in 122 (43.3%), while 90 patients (31.9%) received a D2 and/or a D3. In 110 cases (39.0%), additional resections of nearby organs directly infiltrated by the tumor (T4b) were carried out (Table 2).

As regards the treatment of metastases, hepatic resection was performed in 37 (13.1%) cases; in 82 patients (29.1%), a resection of distant lymph nodes (stations 12, 13, 16 and occasionally more distant stations) was carried out; peritonectomy was performed in 105 (37.2%) cases. In 49 cases, peritonectomy was completed by HIPEC.

Radical resection (R0) was achieved in 117 cases (41.5%), while a microscopically (R1) non-curative resection or a palliative R2 gastrectomy was performed in 75 and 90 patients, respectively.

Considering only the 209 patients submitted to surgery with curative intent (preoperative planning of both gastric neoplasm and metastasis resection), a R0 resection rate of 56% was obtained. Unexpected intra-operative up-staging of metastatic disease or technical reasons related to metastasis localization were the factors that led to palliative gastrectomy.

The overall post-operative complication rate was 26.2% (Table 3), and 7 patients died in the postoperative period (2.5%).

Chemotherapy was administered as neoadjuvant treatment in 43 cases (12.2%) and as adjuvant therapy in 69 cases (25%). Forty-six patients (16.3%) received both pre-operative and post-operative chemotherapy, in 45 cases in the setting of conversion treatment [22]. One hundred and thirty-four patients (47.5%) did not receive chemotherapy. Post-operative complications altered the planned therapeutic strategy: 36/74 patients (48%) suffering major complications were not in a condition to receive the planned post-operative chemotherapy.

Pathological staging is reported in Table 1. Two hundred and twenty-seven patients (80.5%) had one or more pathologically confirmed metastases; in 38 cases, metastases were not resected (13.5%), and in 2 cases (0.7%), they were diagnosed within 6 months from surgery and both patients were re-operated with R0 intent. In 15 patients (5.3%), we observed a complete regression of the metastatic tumor after preoperative chemotherapy with post-apoptotic fibrotic tissue infiltrated by histiocytes (ypM0).

### 3.1. Overall Survival

Forty-three patients (15.2%) were alive when follow-up was stopped; 14 of them (4.9%) were disease-free while the remaining 29 developed peritoneal and/or lymphatic and/or haematogenous relapse.

Fourteen patients (4.9%) died due to non-neoplastic causes; 18 (6.4%) were lost at follow-up, and in 33 cases (12%) the cause of death was not recorded. One hundred and sixty-seven patients (59.2%) died due to gastric cancer recurrence (66 patients) or progression (101 patients). Recurrence was peritoneal in 24 cases, haematogenous in 17, lymph-nodal in 6, peritoneal and haematogenous in 2, lymph nodal and haematogenous in 3, and peritoneal, lymph nodal and haematogenous in 2 more cases; details concerning recurrence are unknown in 12 cases.

Excluding the 18 patients lost at follow-up and the 7 patients who died after surgery, 257 cases were available for survival analysis. One, 3, and 5-year Overall Survival (OS) rates after surgery were 45.2%, 20.2% and 11.8%, respectively (Figure 1A), with a median OS of 10.9 months (CI 95% 9.4–12.5).

### 3.2. Survival According to Surgical Curativity

Different survival performances (*p* < 0.001) were observed between patients benefitting from curative R0 resection of the entire neoplastic bulk and those who received non-curative gastrectomy (Figure 1B). Median survival was 14, 8.3 and 7.5 months after R0, R1 and R2 resections, respectively. Similarly, 1, 2, and 3-year survival rate was 55.7%, 28.3% and 19.1%, respectively, after curative surgery, 40.3%, 17.1% and 6.4% after R1 resection, and 32.4%, 8.7% and 4.4% after palliative gastrectomy.

### 3.3. Survival According to Metastasis Site

There were no significant differences in survival performance according to metastasis site (Figure 2): median survival was 11.2, 11.6, 9.8, and 21.4 months for patients with peritoneal, hepatic, lymph-nodal and hematogenous extrahepatic metastases, respectively; it was 7.0 months for patients with more than 1 metastatic site (*p* = 0.835). In all subgroups, long-term survivors were observed.

### 3.4. Prognostic Factors from the Entire Cohort

At univariate analysis, several clinical variables displayed statistical value (*p* < 0.05), as indexed in Table 4. At multivariate analysis, the variables that showed an independent effect upon survival were: the number of dissected lymph nodes, the extension of lymphectomy (D), the N factor of the pTNM, the curativity of the surgical procedure, and tumor histology. A worse prognosis was associated with: a low number of resected lymph nodes, D1 lymphadenectomy, pN3 tumors, non-curative surgery, and non-intestinal histology.

### 3.5. Analysis on the Subgroup of R0 Patients

We also performed a survival analysis in the subgroup of 117 patients treated with curative surgery: median OS was 14.0 months (CI 95% 8.8–19.2). One, 3, and 5-year OS rates after surgery were 55.7%, 28.3% and 19.1%, respectively (Figure 1B). Again, no difference was observed according to metastasis site.

Amongst the different variables displaying a significant value at univariate survival analysis, we observed that the variables pT and pN of the gastric primary and extension of lymphectomy (D) displayed an independent prognostic role at multivariate analyses, with a worse prognosis associated with pT>2, pN3a-b and D1 lymphadenectomy (Table 5).

Furthermore, these factors showed a cumulative effect, highlighting that patients with no more than 1 risk factor can reach a median survival of 33 months (Figure 3).

## 4. Discussion

This is a retrospective analysis of a large cohort of Western patients whose stage IV gastric cancer was managed surgically. Our analysis produced intriguing results, which deserve discussion. Three of them appear particularly significant: the clear difference in survival performance between curative R0 resection and palliative gastrectomy, the fact that patient survival was not influenced by metastatic site, and the observation that metastasis-related variables had no impact upon survival.

### 4.1. Survival Performance after R0 or Palliative Surgery

The clear survival advantage of curative R0 surgery over palliative R2 procedures is confirmation that the possibility exists, at least for a selected subgroup of stage IV gastric cancer patients, for integrated management including curative surgery. Based upon our findings, we consider that the REGATTA Trial [5] maintains its value for cases not candidates for curative R0 surgery, yet its conclusions should be discussed when R0 resection is deemed possible and a new trial should perhaps be designed with three study arms, including curative surgery. It is worth noting, however, that R1 and R2 patients displayed similar survival performances. In our eyes, this raises a critical point: the ability to identify the patients that will receive a curative resection from those who will not, employing all possible diagnostic strategies, including redoing diagnostic laparoscopy after preoperative chemotherapy. From this point of view, our performance was poor: an R1 resection rate of 36% (75/209) was recorded, which seems rather high, especially if one considers that these patients had been exposed to the risks of major surgery without benefits and that they would have been the best candidates for chemotherapy.

At first glance, our survival results may appear modest, particularly if compared to those from Eastern series [23,24,25]. However, it should be considered that we treated Western Caucasian patients, who cannot benefit from S1 chemotherapy regimens. Moreover, intestinal histology was only present in 36% of cases and our patients received suboptimal chemotherapy regimens, since the FLOT protocol was introduced only in the later phase of the study period. The results of a recent Spanish study [26] are completely in line with ours, and reported a median OS from surgery including metastasectomy of 16.7 months, which compared favorably to the median OS of 10.4 months of the full series, while 3-year survival was 30.6% versus 8.4% in the overall population. In our series, median OS was 10.9 months and the 3-year survival rate after surgery was 19.8%, yet in the subgroup of R0 patients, the median OS was 14 months and the 3-year survival rate was 26.8%, with 5-year survival above 18%.

We believe this to be a demonstration that the possibility for aggressive management also exists for Western patients, and in this context, conversion therapy can play a significant role. It is impossible to extrapolate the real impact of the different multimodal treatments employed in our series: they were heterogeneous (adjuvant chemotherapy, conversion therapy, HIPEC) and employed in limited subgroups of patients. However, special attention should be paid to the 15 patients in whom a complete regression of the metastatic bulk (ypM0) was observed after chemotherapy and surgical resection, as this may lead to significant and unexpected findings.

### 4.2. Metastatic Site Does Not Influence Survival

Our previous studies [6,9,10,11] focused on hepatic metastases and endorsed an aggressive integrated approach including surgery for selected patients, on the theoretical assumption that hepatic metastasis may still characterize a regional and not a systemic disease, and thus display a better prognosis, since the liver plays a “first-filter” role for the portal bloodstream. Unexpectedly, however, expanding our evaluation to the entire cohort of metastatic patients, we did not observe better survival outcomes for patients with hepatic metastases when compared to those presenting metastatic disease in the peritoneal cavity, distant lymph nodes, extra-hepatic haematogenous locations, or any possible association of the above.

These data confirm the role of the variable M of the TNM system as it is generally recognized, which is not to be discussed and will continue to signal the presence of any metastasis effectively. The need to specify the metastatic site (H, P, L) will be of practical use only for clinical research but it will not affect the general classification of gastric cancer.

The analysis of survival curves showed that survival is not influenced by metastasis site and also that relevant 3-year survival rates ranging around 20% can be achieved, independently of the site of the metastasis. We found long survivors in all subgroups of patients yet this occurrence remains sporadic, albeit not negligible. These important concepts are to be highlighted, as they reveal that a chance of effective treatment, if not of cure, should also be given to stage IV patients, instead of submitting them to palliative chemotherapy or supportive care.

### 4.3. Metastasis-Related Variables Had No Impact Upon Survival

In the era of precision medicine and limited resources, the selection of candidates for an aggressive approach to stage IV gastric cancer gains particular relevance. This is even more true in the Western world, where the curve that describes survival after multimodal treatment including surgery suffers a steep drop during the first year: in the present experience, mortality was around 40% after 6 months and reached 60% one year after surgery.

For this reason, we did our utmost to identify simple clinical elements that could orientate the therapeutic approach. However, we were surprised by the fact that in our cohort of patients, survival was unaffected by the considered metastasis-related variables. Indeed, at multivariate analysis, the variables that independently affected survival were the nodal status of the gastric primary and its histological type.

The nodal involvement was particularly significant: not only did the variable N of the pTNM result to be significant with a *p* = 0.003, but also the extension of lymphectomy (D) and the number of retrieved nodes showed *p* < 0.001 and *p* = 0.023, respectively. These results were similar after analyzing the subgroup of patients who had benefitted from a R0 resection both on gastric primary and metastases, with clear independent prognostic value of the extension of lymphectomy and of the variable N. In this subgroup, we also noted the prognostic role of the variable T of the pTNM. The prognostic roles of variables T and N were expected, as they had constantly emerged in our previous works addressing hepatic metastases from gastric cancer. They also displayed a cumulative prognostic effect (Figure 3), which clearly shows the importance of extended lymphectomy and suggests surgical abstention, if possible, for those unfit for this kind of procedure.

From the cultural point of view, the lack of correspondence between the different possible declinations of the variable M+ and worsening of the prognosis somehow subverts our expectations and confirms the classical interpretation of the TNM, at least in the subset of metastatic patients who are treated surgically. On the contrary, a major prognostic role is played by the variable N, which we considered hierarchically inferior in suggesting poor survival compared to the variable M.

The prognostic role of the lymphatic sphere in stage IV gastric cancer underlines and recalls the role of surgical technique and the importance of high-quality surgery on the gastric primary, even in the presence of metastatic spread, with the removal of the entire metastatic bulk. We did not expect that the diffusion of aggressive and possibly integrated therapeutic approaches would promote the surgeon’s role in the same subset of patients for whom the general approach is to adopt the simplest surgical solution, since the stage of the disease does not allow appreciable survival. It should be noted that lymphectomy plays a therapeutic role, as we clearly show an increase in median survival parallel to the increase in the number of retrieved nodes. Furthermore, the surgeon must display a certain degree of eclecticism as adjunctive resections of nearby organs are often required in this particular subset of patients (39% of cases in our series, Table 2).

The importance of high-quality surgery is further enhanced by the clear independent prognostic role (*p* = 0.032) played by curative R0 surgery, achieved both on gastric primary and on metastasis. R0 resection is more likely to be achieved in patients with good Performance Status not requiring surgery in emergency conditions, and it must be pursued in referral centers by surgeons who strictly adhere closely to common-sense guided surgical principles, capable of guaranteeing extremely low mortality and morbidity rates. Indeed, the increase in the biologic cost of these procedures may easily hamper all possible efforts to improve patient prognosis.

## 5. Limitations of the Study

This work depicts a subset of surgically managed stage IV gastric cancer patients as they appear in everyday clinical practice. However, it suffers from all the limitations originating from its retrospective and non-comparative nature. This was particularly evident when considering the chemotherapy protocols, which changed during the study period and among the different centers, or when considering indications to HIPEC, which was not employed in all participating centers. In particular, our study suffers from its surgical nature and the fact that our cohort is not representative of stage IV gastric cancer patients. In general terms, our cohort had a double selection: only those patients in the best general condition reached the surgical theater, and curative surgery was offered to the best candidates from both the performance status point of view and also from the oncologic.

For this reason, our group has already started a prospective registry with the aim of evaluating this controversial topic more thoroughly.

## 6. Conclusions

Our data demonstrate that for stage IV gastric cancer patients, the possibility for effective care exists, at least for a subgroup of them. Each case should be discussed in a multidisciplinary context in order to identify the patients who may benefit from an aggressive integrated approach, which opens the door to interesting and unexpected mid- and, possibly, long-term results, independently of the site of metastases, especially if surgery with curative intent can be pursued.

## Figures and Tables

**Figure 1 cancers-12-00158-f001:**
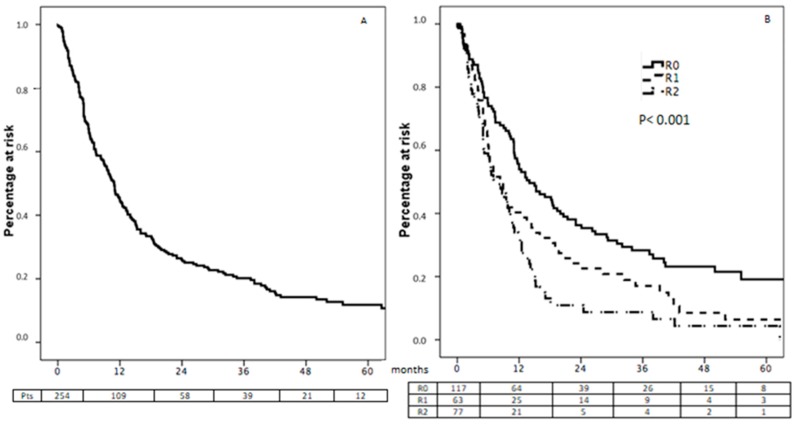
Overall survival in the total population (**A**) and according to the curativity of surgical resection (**B**).

**Figure 2 cancers-12-00158-f002:**
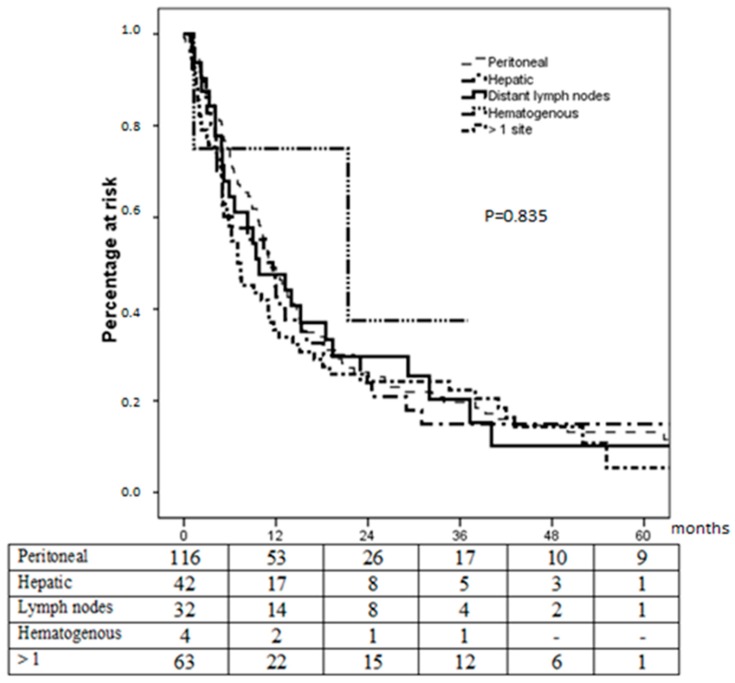
Survival according to metastases site.

**Figure 3 cancers-12-00158-f003:**
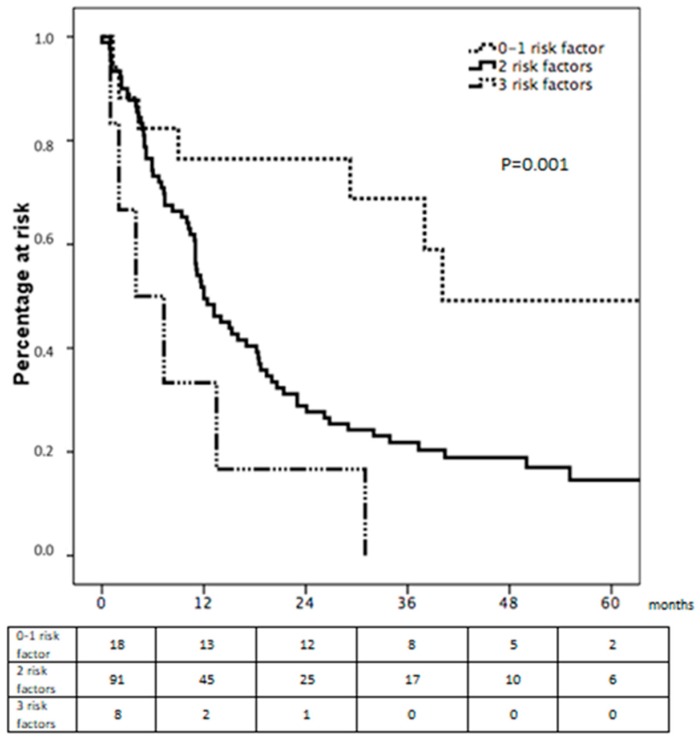
Cumulative effect of negative prognostic factors.

**Table 1 cancers-12-00158-t001:** Overall population characteristics: clinical and pathological TNM.

Clinical Staging (cTNM)			Pathological Staging (pTNM)		
**cT**	**n**	**%**	**pT**	**n**	**%**
**1**	2	0.7	**1**	1	0.3
**2**	13	4.6	**2**	8	2.8
**3**	87	30.9	**3**	93	33.0
**4**	162	57.4	**4a**	147	52.2
**-**	-	-	**4b**	31	11.0
Unknown	18	6.4	Unknown	2	0.7
**cN**	**n**	**%**	**pN**	**n**	**%**
**0**	54	19.2	**0**	16	5.7
**1**	208	73.7	**1**	26	9.2
**x**	8	2.9	**2**	38	13.5
**-**	-	-	**3a**	92	32.6
**-**	-	-	**3b**	86	30.5
Unknown	12	4.2	Unknown	24	8.5
**cM**	**n**	**%**	**pM**	**n**	**%**
**0**	107	37.9	**0 ***	2	0.7
**1**	175	62.1	**1**	227	80.5
-	-	-	**Unresected metastases**	38	13.4
-	-	-	**ypM0**	15	5.4

* Synchronous metastases diagnosed within 6 months after surgery.

**Table 2 cancers-12-00158-t002:** Additional resections of organs apparently infiltrated by the tumor (cT4b).

Organ	No.	%
Pancreas-spleen	7	2.5
Spleen	21	7.4
Pancreas	13	4.6
Colon	9	3.1
Liver	29	10.3
Small bowel	4	1.4
Other	10	3.5
>1	17	6

**Table 3 cancers-12-00158-t003:** Operative morbidity.

Clavien–Dindo Scale	No.	%
3a	60	21.3
3b	8	2.8
4	6	2.1

**Table 4 cancers-12-00158-t004:** Prognostic factors: univariate and multivariate analyses (257 cases).

Prognostic Variables	No.	Median Overall Survival (OS)	Univariate *p*-Value	Multivariate *p*-Value	Odds Ratio (±CI 95%)
**Performance status (ECOG)**					
0	111	11 (7.5–14.8)	0.029	n.s.	
1–2	35	9 (4.9–13.1)			
Unknown	111				
**Gastric occlusion-perforation-bleeding**					
No	153	11.2 (8.5–13.8)	0.002	n.s.	
Yes	104	9 (6.1–11.9)			
**Lymphadenectomy**					
D1	58	6.6 (4.2–8.9)	<0.001	<0.001	0.5 (0.4–0.8)
D2	116	12.4 (10.0–15.0)			
D3	83	10.9 (6.3–15.5)			
**Curativity**					
R0	117	14.0 (8.8–19.2)	<0.001	0.032	1.6 (1.1–2.3)
R1	63	8.3 (5.2–11.4)			
R2	77	7.5 (4.4–10.6)			
**Gastric margin**					
Negative	202	12.0 (10.1–13.8)	<0.001	n.s.	
Positive	55	5.4 (4.2–6.6)			
**pT**					
1–2	8	30.8 (15.9–54.1)	<0.001	n.s	
3	83	11.2 (6.8–15.5)			
4a	139	10.7 (8.8–12.5)			
4b	27	5 (4.1–5.9)			
**pN**					
0	16	38 (10.9–65.1)	<0.001	0.003	1.9 (0.9–4.0)
1	24	23.9 (0.4–64.4)			
2	34	16.0 (5.0–27.0)			
3a	86	7 (2.5–11.5)			
3b	77	9.5 (6.4–12.5)			
x	20	10.1 (2.1–18.0)			
**Number of resected nodes**					
<16	23	6 (1.1–10.9)	0.041	0.028	0.07 (0.02–0.3)
16–22	42	5.4 (3.3–7.5)			
23–29	57	11.6 (10.2–13)			
30–44	64	12.5 (8.1–16.9)			
45–59	53	10.2 (0.1–27.2)			
60–74	13	15 (0.3–35)			
>74	5	7.4 (0.1–16.2)			
**Histology**					
Intestinal	103	12.5 (10.3–14.6)	0.015	0.023	1.7 (1.1–2.5)
Diffuse	116	9.4 (6.6–12.2)			
Mixed	31	7.0 (5–8.9)			
Other	7	6.7 (5.2–8.2)			

n.b. Variables considered as non-emerging at univariate analysis are listed in Appendix A, Table A1.

**Table 5 cancers-12-00158-t005:** Prognostic factors: univariate and multivariate analyses (R0 cases).

Prognostic Variables	No.	Median OS	Univariate *p*-Value	Multivariate *p*-Value	Odds Ratio (±CI 95%)
**Lymphadenectomy**					
D1	8	7.3 (5.2–9.3)	0.001	0.014	0.4 (0.2–0.8)
D2	58	16.0 (9.8–22.1)			
D > 2	51	14.0 (6.2–21.7)			
**pT**					
1–2	7	38.0 (20.0–82.0)	0.006	0.042	9.4 (1.1–81.6)
3	40	11.2 (5.0–17.4)			
4a	56	18.7 (9.7–27.6)			
4b	14	5.0 (1.7–8.3)			
**pN**					
0	14	38.0 (26.5–49.5)	0.001	0.016	3.0 (1.3–7.1)
1	10	82.4 (25.0–85.0)			
2	21	21.4 (14.5–28.2)			
3a	36	11.0 (6.1–15.9)			
3b	32	15.0 (9.5–20.5)			
x	4	6.0 (4.1–7.9)			
**Number of resected nodes**					
<16	5	6.0 (1.8–10.1)	0.034	n.s.	
16–22	17	5.0 (0.1–18.5)			
23–29	25	15.3 (9.3–21.3)			
30–44	36	18.4 (8.0–28.8)			
45–59	26	14.0 (2.7–25.2)			
60–74	6	21.4 (0.2–53.0)			
>74	2	1.3			
**Hystology**					
Intestinal	49	13.5 (5.9–21.0)	0.011	n.s.	
Diffuse	52	16.0 (10.7–21.2)			
Mixed	11	7.3 (0.8–13.7)			
Other	5	15.3 (0.1–33.7)			
**Chemotherapy**					
Yes	77	17.0 (9.8–24.2)	0.040	n.s.	
No	40	11.0 (3.8–18.1)			

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
