# Peer review of "Stage IV Gastric Cancer: The Surgical Perspective of the Italian Research Group on Gastric Cancer"

_cancers, 2020, doi:10.3390/cancers12010158_

Round 1

Reviewer 1 Report

Thank you very much for giving me a good opportunity to review your article. You described “Stage 4 Gastric Cancer. The Surgical Perspective of the Italian Research Group on Gastric Cancer” It’s very interesting, however, you did not answer primary problem in Stage 4 gastric cancer in your manuscript.

The primary problem for Stage 4 gastric cancer is which is best treatment, Chemotherapy Palliative gastrectomy Radical gastrectomy with resection of metastatic site

You mean R2 is palliative gastrectomy? R1 is gastric margin positive who underwent radical gastrectomy with resection of metastatic site?

It is very confusing, and you should clearly divide palliative gastrectomy with radical gastrectomy with resection of metastatic site.

Patients that pT1-2 and/or pN0 who received at least a D2 lymphectomy were very few, and rarely useful.

Figure 3 is meaningless because D>2(0r.f.) and D1(1r.f.) is only 1 patient.

Author Response

Response to reviewer 1

Thank you very much for giving me a good opportunity to review your article. You described “Stage 4 Gastric Cancer. The Surgical Perspective of the Italian Research Group on Gastric Cancer” It’s very interesting, however, you did not answer primary problem in Stage 4 gastric cancer in your manuscript.

Thank you for your comments and for the time you dedicated for suggestions on which basis we greatly improved our manuscript which is now much better than before.

The primary problem for Stage 4 gastric cancer is which is best treatment, Chemotherapy Palliative gastrectomy Radical gastrectomy with resection of metastatic site

We resolved the primary problem you highlighted, concerning the pest possible treatment for our patients. In fact we dedicated new paragraph to the comparison of survival on the basis of the different R status.

You mean R2 is palliative gastrectomy? R1 is gastric margin positive who underwent radical gastrectomy with resection of metastatic site?

We developed the “material and methods” section in order to remove any possible confusion concerning the status R0-1-2 of resection.

It is very confusing, and you should clearly divide palliative gastrectomy with radical gastrectomy with resection of metastatic site.

We changed figure 1b and introduced survival curves according to the R status so to clearly identify the different subgroups.

Patients that pT1-2 and/or pN0 who received at least a D2 lymphectomy were very few, and rarely useful.

We agree that pt1/2 and/or N0 patients are few, but they exist. In any case we modified figure 3 in order to enhance its role.

Figure 3 is meaningless because D>2(0r.f.) and D1(1r.f.) is only 1 patient.

You will find major modifications requested by you and by the 2 other reviewers enhanced in grey in the text.

Reviewer 2 Report

This was a study trying to clarify that there is/are difference(s) in all stage IV gastric cancer and that surgical treatment may bring benefit to selective stage IV patients.

This study no doubt dealt with an interesting issue of gastric cancer and may have impact on surgical practice of metastatic gastric cancer. However, my suggestion for this paper is reject due to the following comments.

Overall, the included patient population is too heterogeneous to make a solid conclusion The median overall survival of the entire cohort is not better than the historical data, and not surprisingly in patients with R0 resection, the median overall survival reached 14 months. Regarding the prognostic analysis, the number and stratification of patients for prognostic analysis is confusing. The sum of the number in each parameter is not 258. The analysis is questionable due to too many missing data. What is the definition of R0 study in this study? For non-metastatic disease, the status of resection (R0, R1, or R2) can be simply obtained by surgical specimen (stomach). Authors should clarify the definition or mention this problem as a limitation. In table 5, there were 8 patients of R0 resection with “D1 dissection”. All these patients should be operated with curative intent. D1 dissection is not logical under this circumstance. In this study, there were several confounding factors interfering the result, such as application of HIPEC, regimens of chemotherapy, and biological factors of tumor. What are the indications for patients submitted to receive hyperthermic intraperitoneal chemotherapy? What are the regimens? Author should clearly illustrate the limitation of this study instead of just emphasize on difficulty of conduction prospective trial for this issue. Surgical complications should be mentioned clearly which may influence the timing of post-op chemotherapy.

Author Response

This was a study trying to clarify that there is/are difference(s) in all stage IV gastric cancer and that surgical treatment may bring benefit to selective stage IV patients.

This study no doubt dealt with an interesting issue of gastric cancer and may have impact on surgical practice of metastatic gastric cancer. However, my suggestion for this paper is reject due to the following comments.

Overall, the included patient population is too heterogeneous to make a solid conclusion The median overall survival of the entire cohort is not better than the historical data, and not surprisingly in patients with R0 resection, the median overall survival reached 14 months. Regarding the prognostic analysis, the number and stratification of patients for prognostic analysis is confusing. The sum of the number in each parameter is not 258. The analysis is questionable due to too many missing data. What is the definition of R0 study in this study? For non-metastatic disease, the status of resection (R0, R1, or R2) can be simply obtained by surgical specimen (stomach). Authors should clarify the definition or mention this problem as a limitation. In table 5, there were 8 patients of R0 resection with “D1 dissection”. All these patients should be operated with curative intent. D1 dissection is not logical under this circumstance. In this study, there were several confounding factors interfering the result, such as application of HIPEC, regimens of chemotherapy, and biological factors of tumor. What are the indications for patients submitted to receive hyperthermic intraperitoneal chemotherapy? What are the regimens? Author should clearly illustrate the limitation of this study instead of just emphasize on difficulty of conduction prospective trial for this issue. Surgical complications should be mentioned clearly which may influence the timing of post-op chemotherapy.

Thank you for your comments and for the time you dedicated for suggestions on which basis we greatly improved our manuscript which is now much better than before.

1) in the “Limits of the study” paragraph we acknowledged the biases you refer to;

2) We agree that our survival results are not better than the historical data but we consider that a median survival of 14 months after R0 resection merit attention as it was achieved in western stage IV gastric cancer patients.

3) We controlled all numbers and statistics. Prognostic analysis included 257 patients. The sum of the number in each parameter in table 4 (prognostic analysis) is 257 in all cases except for the performance status, because in this case we have a high percentage of missing value; we checked the number reported in the table 4 and we corrected 2 errors. Unfortunately due to the multicentric retrospective nature of the study a certain number of missing data is inevitable and it concerns especially some clinical data and tumor characteristics, while the pathological data are more complete. For this reason we do not believe that the statistical analysis is questionable.

4) We developed the “material and methods” section in order to remove any possible confusion concerning the status R0-1-2 of resection.

5) Concerning the appropriateness of D1 dissection in stage IV gastric cancer we have no basis to judge it, because there are no guidelines on the subject. Our feeling is coincident with yours. In any case, those 8 patients received a D1 lymphectomy because the surgeon choose this strategy. These cases are instrumental in demonstrating that at least a standard D2 dissection must be performed in order to achieve a real R0.

6) We developed the “material and methods” section and introduced chemotherapy protocols and summary HIPEC description.

7) We mentioned the impact of surgical complication on the pre-established therapeutic strategy; unfortunately we are not able in such a short notice to enumerate all the different complications

You will find major modifications requested by you and by the 2 other reviewer enhanced in grey in the text.

Reviewer 3 Report

The demographics of the sites of metastatic disease are informative, and it is notable that only 26% of patients had multiple sites of metastatic disease.  Similar to other reports, the authors note that the peritoneum is the most common site of metastatic disease.

The authors appear to include a patient with a synchronous metastasis diagnosed within 6 months of surgery.  Was surgery performed again? If not, these 2 patients do not fall within the stated purpose of the study. The inclusion of these patients in table 1 is confusing.

What percentage of patients never received chemotherapy?

5% of patients were alive without disease, which is the most important finding of the study.

The authors have not acknowledged the most important limitation of the study:  They may have selected patients with limited disease, good functional status, and an excellent response to chemotherapy.  These patients may do just as well without surgery.

The study is hypothesis generating and could help inform future trials of surgery in stage IV gastric cancer.  However, the authors must appropriately acknowledge the clear selection bias of this retrospective, non-comparative work.

Author Response

The demographics of the sites of metastatic disease are informative, and it is notable that only 26% of patients had multiple sites of metastatic disease.  Similar to other reports, the authors note that the peritoneum is the most common site of metastatic disease.

The authors appear to include a patient with a synchronous metastasis diagnosed within 6 months of surgery.  Was surgery performed again? If not, these 2 patients do not fall within the stated purpose of the study. The inclusion of these patients in table 1 is confusing.

What percentage of patients never received chemotherapy?

5% of patients were alive without disease, which is the most important finding of the study.

The authors have not acknowledged the most important limitation of the study:  They may have selected patients with limited disease, good functional status, and an excellent response to chemotherapy.  These patients may do just as well without surgery.

The study is hypothesis generating and could help inform future trials of surgery in stage IV gastric cancer.  However, the authors must appropriately acknowledge the clear selection bias of this retrospective, non-comparative work.

Thank you for your comments and for the time you dedicated for suggestions on which basis we greatly improved our manuscript which is now much better than before.

We explained in the text that both patients with metastasis diagnosed within 6 months from surgery had been re-operated on with curative intent. 5% of patients did receive chemotherapy. We introduced this data in the text. We developed the “Limits of the study” paragraph and we acknowledged the biases you refer to;

You will find major modifications requested by you and by the 2 other reviewer enhanced in grey in the text.

Round 2

Reviewer 1 Report

Your revised manuscript is correctly revised, and should be accepted.

Author Response

Comments and Suggestions for Authors

Your revised manuscript is correctly revised, and should be accepted.

Dear Reviewer,

We have revised the English language of the article and corrected a few errors concerning the numbers in the text which had suffered the “copy and paste” syndrome.

Thank you for your time and constructive contribution.

Reviewer 2 Report

First of all, my suggestions are still reject because Cancers is a high impacted journal. The authors have made a major revision, however, it is still hard to read and get confused. Especially the sum of the patient number is always not 282, including form the very early beginning part of the abstract through the whole text. Most importantly, the sum of the number is different in every table to means the analysis is questionable due to too much missing data. This is a key problem of this retrospective, multi-center study and hard to be solved at all unless this is a prospective study.

Author Response

Comments and Suggestions for Authors

First of all, my suggestions are still reject because Cancers is a high impacted journal. The authors have made a major revision, however, it is still hard to read and get confused. Especially the sum of the patient number is always not 282, including form the very early beginning part of the abstract through the whole text. Most importantly, the sum of the number is different in every table to means the analysis is questionable due to too much missing data. This is a key problem of this retrospective, multi-center study and hard to be solved at all unless this is a prospective study.

Dear Reviewer,

We have revised the English language of the article and corrected a few errors concerning the numbers in the text which had suffered the “copy and paste” syndrome. The total of patients in the work was 282 and on this number we analyzed operative results. Survival analysis was conducted on 257 cases as 7 patients died after surgery and 18 were lost at follow-up. All major variables are fully available for analysis. Indeed only ECOG status and ASA score registered a good amount of missing data.

Please note the revisions marked in green and check the figures.

Unfortunately we can not change the retrospective nature of this work, which carries a limits but also a good amount of value.

Thank you for your time and constructive contribution.

This manuscript is a resubmission of an earlier submission. The following is a list of the peer review reports and author responses from that submission.